# TRIATODEX, an electronic identification key to the Triatominae (Hemiptera: Reduviidae), vectors of Chagas disease: Development, description, and performance

Rodrigo Gurgel-Gonçalves[1,2☯]*, Fernando Abad-Franch[2☯], Maxwell Ramos de Almeida[3], Marcos Takashi Obara[2,4], Rita de Cássia Moreira de Souza[5], Jainaine Abrantes de Sena Batista[6], Douglas de Almeida Rocha[1,2]

**1** Laboratório de Parasitologia Médica e Biologia de Vetores, Área de Patologia, Faculdade de Medicina, Universidade de Brasília, Brasília, Brazil, **2** Faculdade de Medicina, Núcleo de Medicina Tropical, Universidade de Brasília, Brasília, Brazil, **3** Real Comércio e Indústria de Alumínio Ltd., Brasília, Brazil, **4** Faculdade de Ceilândia, Universidade de Brasília, Brasília, Brazil, **5** Grupo Triatomíneos, Instituto René Rachou–Fiocruz Minas, Belo Horizonte, Brazil, **6** H2J Comunicação & Marketing, Brasília, Brazil

☯ These authors contributed equally to this work.
* gurgelrg@hotmail.com

## Abstract

Correct identification of triatomine bugs is crucial for Chagas disease surveillance, yet available taxonomic keys are outdated, incomplete, or both. Here we present TRIATODEX, an Android app-based pictorial, annotated, polytomous key to the Triatominae. TRIATODEX was developed using Android Studio and tested by 27 Brazilian users. Each user received a box with pinned, number-labeled, adult triatomines (33 species in total) and was asked to identify each bug to the species level. We used generalized linear mixed models (with user- and species-ID random effects) and information-theoretic model evaluation/averaging to investigate TRIATODEX performance. TRIATODEX encompasses 79 questions and 554 images of the 150 triatomine-bug species described worldwide up to 2017. TRIATODEX-based identification was correct in 78.9% of 824 tasks. TRIATODEX performed better in the hands of trained taxonomists (93.3% *vs.* 72.7% correct identifications; model-averaged, adjusted odds ratio 5.96, 95% confidence interval [CI] 3.09–11.48). In contrast, user age, gender, primary job (including academic research/teaching or disease surveillance), workplace (including universities, a reference laboratory for triatomine-bug taxonomy, or disease-surveillance units), and basic training (from high school to biology) all had negligible effects on TRIATODEX performance. Our analyses also suggest that, as TRIATODEX results accrue to cover more taxa, they may help pinpoint triatomine-bug species that are consistently harder (than average) to identify. In a pilot comparison with a standard, printed key (370 tasks by seven users), TRIATODEX performed similarly (84.5% correct assignments, CI 68.9–94.0%), but identification was 32.8% (CI 24.7–40.1%) faster on average–for a mean absolute saving of ~2.3 minutes per bug-identification task. TRIATODEX holds much promise as a handy, flexible, and reliable tool for triatomine-bug identification; an updated iOS/Android version is under development. We expect that, with continuous refinement derived from evolving knowledge and user

**Data Availability Statement:** All relevant data are within the manuscript and its Supporting Information files.

**Funding:** DAR received specific funding of Coordenação de Aperfeiçoamento de Pessoal de Nível Superior https://www.capes.gov.br/ Award Number: finance code 001. RGG received specific funding of Conselho Nacional de Desenvolvimento Científico e Tecnológico http://www.cnpq.br/ Award Number: 426619/2018-8. JASB received salary from H2J Comunicação & Marketing, Brasília, Brazil and MRA received salary from Real Comércio e Indústria de Alumínio Ltd. The specific roles of these authors are articulated in the 'author contributions' section. The funders did not have any role in the study design, data collection and analysis, decision to publish, or preparation of the manuscript.

**Competing interests:** JASB received salary from H2J Comunicação & Marketing, Brasília, Brazil and MRA received salary from Real Comércio e Indústria de Alumínio Ltd. This does not alter our adherence to PLOS ONE policies on sharing data and materials. There are no patents, products in development or marketed products to declare.

feedback, TriatoDex will substantially help strengthen both entomological surveillance and research on Chagas disease vectors.

## Introduction

The correct identification of pathogen vectors is critical both for understanding disease dynamics and for effective disease control and surveillance. Vector identification has traditionally relied on printed dichotomous keys based on morphological characters (e.g., [1–3]). Today, however, standard printed keys are being superseded by electronic alternatives [4]. Among other appealing features, electronic keys are easy to update (with, e.g., new taxa or distribution records) and enhance (e.g., by improving character or character-state descriptions and illustrations); can run on hand-held devices including smartphones; and can link users to rich additional resources via the World Wide Web. Electronic keys are available for, e.g., sand-flies, biting midges, mosquitoes, and tsetse flies [5–9]. TriatoKey [10] is an electronic key to 42 triatomine-bug species recorded in Brazil; it was developed to support identification by non-specialist disease-surveillance and community-health technicians [10]. Electronic keys based on smartphone technology should be especially helpful for identifying vectors in the field. This may help enhance surveillance by professional staff and is also opening disease-vector surveillance to citizen science [11–13].

Enhanced surveillance is particularly important for vector-borne diseases that are widely distributed over remote rural areas; for which entomological and epidemiological data are sparse or even locally unavailable; and that are transmitted by many vector species, which complicates vector identification and limits our basic knowledge about many of the rarer species. Chagas disease, one of the major neglected tropical diseases, has all these characteristics [14,15].

Chagas disease is caused by infection with *Trypanosoma cruzi*, a protozoan parasite endemic to the Americas and primarily transmitted among mammals including humans by blood-sucking triatomine bugs–of which 150+ species are recognized at present [1,14–16]. The World Health Organization (WHO) and the Global Burden of Diseases Study (GBD) estimate that 5–7 million people carry the parasite worldwide, mainly in Latin America [17,18]. Extensive vector-control campaigns nearly eliminated house infestation by two non-native triatomine-bug species in large portions of southern South America (*Triatoma infestans*) and Central America (*Rhodnius prolixus*) [19,20]. However, incidence estimates by the WHO (~40,000 new infections in 2010 [17]) and the GBD (~160,000 new infections in 2017 [18]) suggest that *T. cruzi* transmission is by no means under control. In most (perhaps ~75–80%) of these cases, transmission is mediated by bugs in any of the 100+ native species that infest or invade houses from the southern United States to southern Argentina [16,21–23].

Although too often underappreciated, this state of affairs clearly calls for a renewed emphasis on entomological surveillance [21,22]. In many settings, surveillance used to focus on one 'primary' and a few 'secondary' vector species, but now needs to deal with a host of native species with interconnected wild and domestic-peridomestic populations [21,22,24,25]. In Brazil, for example, one non-native species, *T. infestans*, was for decades the primary target of entomological surveillance–with some attention directed to the native *Panstrongylus megistus*, *T. brasiliensis*, *T. sordida*, and *T. pseudomaculata* [24–28]. Today, however, surveillance systems routinely record further native species including *T. vitticeps*, *T. tibiamaculata*, *T. maculata*, *T. costalimai*, *T. rubrovaria*, *R. pictipes*, *R. robustus*, *R. nasutus*, *R. neglectus*, *P. lutzi*, or *P.*

*geniculatus*–and several others are occasionally found inside houses [29–33]. Particularly in species-rich regions, these native triatomines pose a dual challenge to Chagas disease surveillance. First, surveillance staff faces the taxonomic challenge of accurately identifying many rare, little-studied species–some of which may be absent from outdated printed keys. Second, surveillance staff and academic researchers both confront the challenge of working out the relative importance of each bug species with very limited empirical data on basic matters such as species distributions or the frequency of house invasion and infestation by different species (e.g., [29,31]).

To help address these two challenges, we have developed a relatively simple, pictorial, annotated, polytomous electronic key to adult triatomine bugs. In its current version, the key, called TRIATODEX, runs on an Android app and covers the 150 triatomine-bug species formally described worldwide up to 2017 (S1 Table). It thus overcomes the taxonomic-coverage limitations of available printed [1,2,34] and electronic [10] keys. Further, we have assessed TRIATODEX performance in a series of blind identification tasks (33 triatomine-bug species in total) completed by 27 volunteer users with different training and professional backgrounds. A pilot trial allowed us to also compare the time needed to complete an identification task with either TRIATODEX or a printed key.

## Materials and methods

### TRIATODEX development

We obtained digital pictures of adult triatomine bugs (dorsal view) from the digital triatomine collection of the Virtual Vector Laboratory [35] and the archives of RG-G, FA-F, and, with permission, Cleber Galvão, James S Patterson, and José M Ayala. TRIATODEX also includes coarse species-distribution maps based on published records [1,29,34,36–39] and brief notes on the ecology and medical relevance of each species.

TRIATODEX's pictorial key is broadly based on the printed dichotomous keys by Lent and Wygodzinsky [1] and Galvão and Dale [2], from which we selected what we considered to be the most critical information for the differentiation of species and higher systematic groupings. We organized the database in Microsoft Excel, with a line for each species and columns containing information on: author(s) and year of species description; overall body length (in mm); coarse geographic distribution; known or potential medical relevance (broadly indexed by whether the species has been recorded infesting or invading human dwellings); known key habitats; and 79 morphological characters. A few species (see below) belong in groups of sibling taxa that cannot be distinguished by external morphological characters such as those included in TRIATODEX (or printed keys); in these cases, the app displays a list of sibling 'candidate species' in the last step of the identification task.

TRIATODEX was built drawing primarily on our previous experience with LUTZODEX, an app for the identification of Brazilian sandfly species [5]. Briefly, we used Android Studio v. 1.5.1 and a library developed by Google with Java v.8. The approach relies on building a Microsoft Excel database separately from the app and then loading the information during program execution by connecting structures and images through the use of tags added to the database [5]. TRIATODEX code is available at https://github.com/eumaxwell/TriatoDex/tree/master/java/dextaxonomia/com/triatodex.

### TRIATODEX performance

TRIATODEX was tested by 27 Brazilian volunteer users, 13 with and 14 without specialized training in triatomine-bug taxonomy: five health-surveillance agents, four laboratory technicians, nine undergraduate students, four graduate students, and five research scientists (Table 1).

**Table 1. TRIATODEX volunteer users: User characteristics and number of identification tasks completed in the assessment of TRIATODEX performance.**

| User ID | Specialized training[a] | Reference lab[b] | Training | Primary job[c] | Age class | Gender | Tasks |
|---------|------------------------|------------------|----------|----------------|-----------|--------|-------|
| U_1 | Yes | No | Biomedicine | Lab technician | 30s | Female | 33 |
| U_2 | Yes | No | Biology | Lab technician | 30s | Female | 33 |
| U_3 | No | No | Biology | Undergraduate | 20s | Male | 33 |
| U_4 | Yes | No | Biology | Surveillance | 20s | Female | 33 |
| U_5 | No | No | High school | Surveillance | 50s | Female | 33 |
| U_6 | No | No | Veterinary | Undergraduate | 20s | Female | 33 |
| U_7 | Yes | No | Biology | Surveillance | 30s | Female | 33 |
| U_8 | Yes | No | Biology | Surveillance | 30s | Male | 33 |
| U_9 | Yes | No | High school | Surveillance | 50s | Male | 33 |
| U_10[d] | Yes | Yes | Biology | Researcher | 40s | Female | 26 |
| U_11 | No | No | Biology | Undergraduate | 20s | Female | 33 |
| U_12[d] | No | Yes | Biomedicine | Lab technician | 30s | Female | 26 |
| U_13[d] | No | Yes | Biology | Graduate student | 20s | Female | 26 |
| U_14 | No | No | Nursery | Undergraduate | < 20 | Female | 33 |
| U_15[d] | No | Yes | Biology | Undergraduate | 20s | Male | 26 |
| U_16[d] | Yes | Yes | Biology | Researcher | 40s | Female | 26 |
| U_17[d] | Yes | Yes | Biology | Graduate student | 20s | Female | 26 |
| U_18[d] | No | No | Biology | Undergraduate | < 20 | Female | 33 |
| U_19 | No | No | Biology | Graduate student | 30s | Male | 25 |
| U_20 | No | No | Biology | Undergraduate | 20s | Female | 33 |
| U_21 | Yes | No | Biology | Lab technician | 50s | Male | 23 |
| U_22 | Yes | No | Biology | Researcher | 40s | Male | 33 |
| U_23 | No | No | Biology | Graduate student | 20s | Male | 33 |
| U_24 | Yes | No | Biology | Researcher | 40s | Female | 33 |
| U_25 | No | No | Biomedicine | Undergraduate | 20s | Male | 33 |
| U_26 | No | No | Veterinary | Undergraduate | 20s | Female | 33 |
| U_27 | Yes | Yes | Biology | Researcher | 30s | Female | 26 |

[a] Users who received (or did not receive) specific training in triatomine-bug taxonomy.

[b] Users working (or not) at a national reference laboratory for triatomine-bug taxonomy.

[c] In our main analyses (based on generalized linear mixed models), we focused on getting estimates of TRIATODEX performance for users primarily involved in health surveillance ('Surveillance' *vs*. the rest) and primarily involved in academic research/teaching ('Researcher' plus 'Graduate student' *vs*. the rest).

[d] Users who also participated in the comparison of TRIATODEX with a standard printed key (ref. [2]).

TRIATODEX users worked in six institutions including three universities, two state health-surveillance departments, and a Ministry of Health reference laboratory for triatomine-bug taxonomy. Other user characteristics are given in Table 1. Each user received a box with up to 33 pinned, number-labeled, adult triatomines, each belonging to one species for a total of 33 species (Table 2). The bugs had been identified by two expert taxonomists (RG-G and RdCMdS) using standard keys [1,2,34] and ancillary information (e.g., geographic origin of the bugs or, for phenotypically similar *Rhodnius* spp., morphometric and DNA-sequence data); TRIATODEX users were blinded to any information regarding problem specimens, including these 'reference standard' identifications. All volunteers were briefly instructed on how to use the app (see links to instruction videos in **TRIATODEX availability** below). Users were then asked to assign, based on TRIATODEX, each number-labeled bug to a species; we call each assignment an identification 'task'. The number of tasks completed by each user is shown in Table 1. Task results were then sent to the project's headquarters, where researchers blinded to user identity scored

**Table 2. Triatomine-bug species used in the assessment of TRIATODEX performance.**

| Tribe | Species[a] | Native-domestic[b] | Tasks | Steps[c] |
|---|---|---|---|---|
| Cavernicolini | *Cavernicola lenti* | No | 25 | 6 |
| Triatomini | *Panstrongylus diasi* | No | 25 | 17 |
| | *Panstrongylus geniculatus* | No | 27 | 14 |
| | *Panstrongylus lignarius* | No | 27 | 16 |
| | *Panstrongylus lutzi* | No | 27 | 17 |
| | *Panstrongylus megistus* | Yes | 27 | 15 |
| | *Panstrongylus rufotuberculatus* | No | 18 | 15 |
| | *Triatoma baratai* | No | 25 | 15 |
| | *Triatoma brasiliensis* | Yes | 27 | 15 |
| | *Triatoma carcavalloi* | No | 18 | 19 |
| | *Triatoma costalimai* | No | 27 | 14 |
| | *Triatoma delpontei* | No | 27 | 18 |
| | *Triatoma guazu* | No | 18 | 21 |
| | *Triatoma infestans* | No | 27 | 18 |
| | *Triatoma juazeirensis* | No | 18 | 18 |
| | *Triatoma lenti* | No | 20 | 19 |
| | *Triatoma maculata* | No | 26 | 18 |
| | *Triatoma matogrossensis* | No | 27 | 15 |
| | *Triatoma melanocephala* | No | 20 | 17 |
| | *Triatoma pseudomaculata* | Yes | 26 | 18 |
| | *Triatoma rubrovaria* | Yes | 27 | 19 |
| | *Triatoma sherlocki* | No | 25 | 10 |
| | *Triatoma sordida* | Yes | 27 | 16 |
| | *Triatoma tibiamaculata* | No | 27 | 14 |
| | *Triatoma vitticeps* | No | 27 | 17 |
| Rhodniini | *Psammolestes tertius* | No | 20 | 6 |
| | *Rhodnius domesticus* | No | 27 | 6 |
| | *Rhodnius ecuadoriensis* | No | 27 | 6 |
| | *Rhodnius nasutus* | No | 27 | 9 |
| | *Rhodnius neglectus* | No | 27 | 9 |
| | *Rhodnius pictipes* | No | 27 | 6 |
| | *Rhodnius prolixus* | No | 27 | 8 |
| | *Rhodnius robustus* | No | 27 | 9 |

[a] As determined by expert taxonomists; TRIATODEX users were blinded to this 'reference standard' identification.

[b] This two-level factor distinguished (i) species native to Brazil that are often found infesting houses (coded '1') from (ii) other species, including Brazilian-native species that seldom, if ever, infest houses and species that are not native to Brazil, irrespective of whether they infest houses; TRIATODEX users were blinded to this classification.

[c] Maximum number of TRIATODEX steps the user has to go through to complete each identification task.

assignments as 'correct' (right species-level identification; coded '1') or 'incorrect' (wrong identification or failure to reach species-level identification; both coded '0').

We were primarily interested on overall TRIATODEX performance, measured as the proportion of correct identifications. In addition, we aimed at testing some focal hypotheses (summarized in Table 3) about possible sources of variation in TRIATODEX performance across selected user and target-species traits. We first computed descriptive statistics based on counts and proportions (with score 95% confidence intervals [CI]). We then used bivariate generalized linear mixed models (GLMMs; [40]) with binomial error distribution and logit link-function to

**Table 3. Hypotheses about sources of variation in TRIATODEX performance.**

| Trait group | Trait | Effect | Rationale |
|---|---|---|---|
| Users | Specialized training | Positive | Specific training in triatomine-bug taxonomy should increase the odds of correct identification |
| | Reference laboratory[a] | Positive | Reference-lab workers may be more acquainted with bug taxonomy and more aware of taxonomic methods |
| | Surveillance worker[b] | Uncertain | Should ideally have no effect if the key is to be useful |
| | Academic research/teaching | Positive | Primary involvement in academic research/teaching may increase correct-identification odds |
| | Biologist | Positive | Biologists may be more acquainted with insect morphology and systematics |
| | Age[c] | Uncertain | Younger users ($\leq$ 40 years old) might find it easier to use a mobile app |
| | Gender | None | Should have no effect |
| Bugs | Rhodniini | Negative | Some Rhodniini species may be particularly difficult to differentiate from each other |
| | Native-domestic[d] | Positive | Users may be more acquainted with local species that are more often found infesting houses |

[a] Seven users with different levels of training and experience (see Table 1) worked at the time of the assessment in a national reference lab for triatomine-bug taxonomy.

[b] Five users with different levels of training and experience (see Table 1) worked at the time of the assessment in two state health-surveillance departments.

[c] Grouped into two classes: 'younger' (up to 40 years old) and 'older' (40 years old or more); see Table 1.

[d] This two-level factor distinguished (i) species native to Brazil that are often found infesting houses (coded '1') from (ii) other species (coded '0'), including Brazilian-native species that seldom, if ever, infest houses and species that are not native to Brazil, irrespective of whether they infest houses.

estimate the 'crude' effects of each user- and species-related trait while accounting for the non-independence of identification tasks completed by the same user and identification tasks involving the same bug species; we did this by specifying user- and species-ID as random factors in all our models. These exploratory models provided a preliminary comparison of TRIATODEX performance across groups of users (Table 1) and across triatomine-bug taxa (Table 2). All analyses were done in R 3.6.3 [41] with packages as specified below.

Our inferential analyses focus on the nine variables shown in Table 3. Prior to building our models, we calculated pairwise Pearson correlations and variance inflation factors (VIFs) for the nine fixed-effect variables (*Hmisc* 4.3–1 [42], *car* 3.0–7 [43], and *corrplot* 0.84 [44] R packages); correlations were all $\leq |0.47|$ and VIFs were all $< 2.0$ (S1 Fig), and we therefore decided to include all variables in downstream analyses. We then built, using package *glmmTMB* 1.0.1 [45], a full additive model including our nine variables (plus the user-ID and bug-species-ID random effects). We used package *MuMIn* 1.43.15 [46] to (i) fit all 511 additive models nested within the full model; (ii) assess model performance based on Akaike's information criterion for small samples (AICc); (iii) compute model-averaged slope coefficients and their CIs; and (iv) get a measure of variable importance as the sum of Akaike weights across the 256 models in which each variable was present [47]. We finally used the top-performing (smallest-AICc) model to predict, with the *ggeffects* 0.14.2 package [48], the estimated marginal means (and CIs) of percent correct TRIATODEX-based identification. Data (S1 Data) and code (S1 Code) underlying these analyses are available as Supporting Information.

Finally, seven users (Table 1) were asked to use both TRIATODEX and the printed pictorial key of Galvão and Dale [2] to identify their task specimens (32 species in total) and to record the time (in minutes) taken to complete each task. We used GLMMs (with user- and species-ID random effects) to provide a preliminary comparative assessment of (i) the proportion of correct assignments and (ii) the time to complete each bug-identification task with either key. For proportions, we used logit-binomial models as above. Since the response variable in the time-to-complete-task models ('time models' hereafter) had no zeros, we selected (using AICc scores) the discrete, zero-truncated error distribution that provided the best fit to the data–which, in our case, was the zero-truncated negative binomial distribution ('truncated_nbinom2' in *glmmTMB*; log link-function). The results of four tasks by two users were excluded

from these latter analyses because time measurements were missing; we therefore analyzed 370 tasks– 185 completed with TRIATODEX and 185 with Galvão and Dale's printed key [2]. See S2 Data and S2 Code for the data and code underlying these analyses. We note that all our analyses were done by researchers blinded to TRIATODEX user identity.

### TRIATODEX availability

The app is available (in English and Portuguese) for free download on the Google Play Store (https://play.google.com/store/apps/details?id=max.com.triatodex). TRIATODEX use instructions are also available in Portuguese (https://www.youtube.com/watch?v=8-M55EoWjg0) and English (https://www.youtube.com/watch?v=HGj_wi5DCLA).

## Results

### TRIATODEX description

In its current version (August 2020), which is the one we used in performance trials, TRIATODEX covers the 150 triatomine-bug species described up to 2017 (S1 Table); for each species but *T. gomeznunezi*, the app features a dorsal picture of an adult specimen and a coarse (country- or state/province-level) distribution map. The key includes a total of 79 questions with two to eight possible answers and 254 detail images of morphological structures. TRIATODEX has a main menu (Fig 1) with the following options: (i) 'Search' shows the questions used for identification; (ii) 'Morphological structures' shows the main structures used during identification (head, thorax, legs, abdomen; e.g., Rhodniini leg features in Fig 1); (iii) 'Possible species' shows the list of candidate species, which progressively shrinks from the 150 initial possibilities to one species (or, in a few cases, a short list of sibling species; see below) at the final step of an identification task; (iv) 'Recent answers' shows the list of questions answered up any given

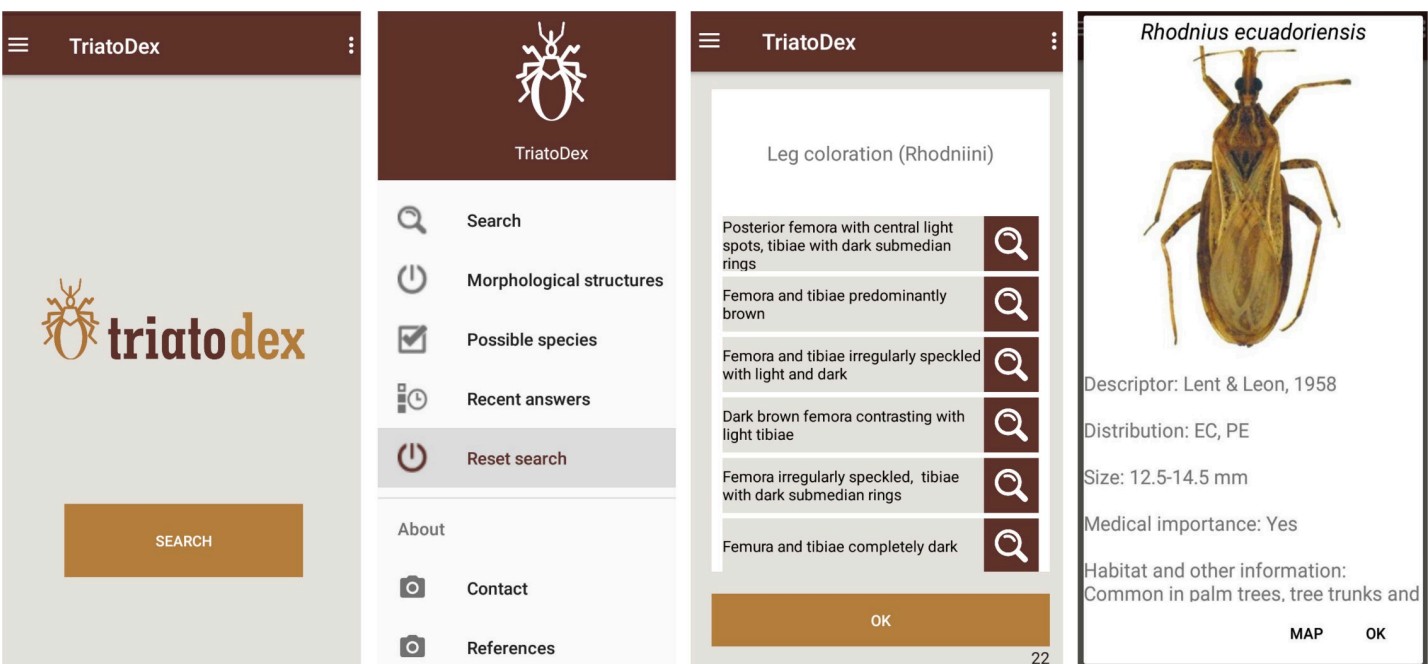

**Fig 1. TRIATODEX screens: Opening screen; main menu; example of questions and answers (note the magnifying-glass icons, which link to detail images of each structure); and example of last step with a picture of an adult specimen (here, *Rhodnius ecuadoriensis*) and taxonomic, distributional (note the link to the map), morphological, and ecological notes.**

point, so that users can 'walk back' the identification process and correct previous answers; (v) 'Reset search' allows users to restart an identification task; (vi) 'Contact' displays the names and e-mail addresses of the developers, so that users can send feedback; and (vii) 'References' shows a list of the main references used to develop the app.

When selecting an answer to the active question, the user is taken directly to the next question, and so on until the identification is completed; then, the app displays the dorsal picture and distribution map of the species, along with the author(s) and year of first description, countries (or, when available, states/provinces) with occurrence records, the body length of adult specimens (in mm), whether it may be considered medically important, and brief notes (when available) about habitats and habits (Fig 1). At each step, the app also displays (i) links (via a magnifying-glass icon; Fig 1) to images with details of the morphological characters each possible answer refers to and (ii) the number (and a link to the list) of candidate species–which, as mentioned above, shrink towards one (or a few) as identification proceeds. From the list of candidate species, users can check information on their distribution, overall size, ecology, and medical importance. The number of identification steps varies from four (for some morphologically distinct *Belminus* species) to 21 (for some *Triatoma* species); on average, it takes 12 steps to complete an identification task.

## TRIATODEX performance

Overall, identification was correct in 78.9% (score CI 76.0–81.5%) of 824 tasks completed by 27 volunteer users. Exploratory bivariate analyses suggested that, as expected (Table 3), TRIATODEX may perform better in the hands of trained specialists (90.3% correct identifications, whereas non-specialists correctly identified 68.6% of the specimens) (Table 4). Conversely, performance appeared to be worse for undergraduate students than for users in other primary-job categories (Table 4). Note, however, that none of the nine undergraduate users had received specialized training in triatomine-bug taxonomy (Table 1); to provide a test of our hypothesis on specialized-training effects (see Table 3), we used this latter variable in downstream multivariate modeling (Table 4). The results of other exploratory comparisons are presented in Table 4; they suggest little to no effects of gender, age, basic training, or type of institution on TRIATODEX performance.

In multivariate analyses, both model-averaged adjusted estimates (Table 5) and the smallest-AICc model within our model set (S2 Table) revealed a clear positive effect of specialized taxonomic training on the odds of correct bug identification with TRIATODEX. In contrast, such odds varied little with user age-class, gender, or basic training in biology; further, whether the user's primary job involved academic research/teaching or disease surveillance, or whether the user worked in research laboratories (including a reference laboratory for triatomine-bug taxonomy) or in vector-surveillance departments had similarly negligible effects on TRIATODEX performance (Table 5). Bug-species traits hypothesized to possibly affect identification odds (Table 3) were also unimportant and had no measurable effects (Table 5). Model-averaged predictions across TRIATODEX users and test species are presented in Figs 2 and 3; the figures emphasize how users with specialized training in triatomine-bug taxonomy were more likely to correctly identify bugs (Fig 2) across our sample of 33 species, including those that were overall harder to identify (Fig 3).

The top-performing GLMM in our model set had a single fixed-effect predictor–whether the user had/had not received specialized training in triatomine-bug taxonomy (S2 Table). This model predicts that, on average, TRIATODEX users with such training would correctly complete 93.25% (CI 88.57–96.10) of identification tasks, *vs*. 72.66% (CI 62.31–81.04) for users without such training. In addition, the top-ranking GLMM suggests that random variation in

**Table 4. TRIATODEX performance across user traits: Descriptive results and exploratory data analyses.**

| User trait | Users | Tasks | Correct | % | CI-low | CI-up | Odds ratio[a] | CI-low | CI-up |
|---|---|---|---|---|---|---|---|---|---|
| Gender[b] | | | | | | | | | |
| Female | 18 | 552 | 429 | 77.7 | 74.1 | 81.0 | Reference | | |
| Male | 9 | 272 | 221 | 81.3 | 76.2 | 85.4 | 1.259 | 0.524 | 3.030 |
| Age[b] | | | | | | | | | |
| Up to 40 | 20 | 617 | 477 | 77.3 | 73.8 | 80.4 | Reference | | |
| Over 40 | 7 | 207 | 173 | 83.6 | 77.9 | 88.0 | 1.692 | 0.657 | 4.354 |
| Specialized training[b] | | | | | | | | | |
| No | 14 | 433 | 297 | 68.6 | 64.1 | 72.8 | Reference | | |
| Yes | 13 | 391 | 353 | 90.3 | 86.9 | 92.8 | **5.201** | **2.930** | **9.230** |
| Primary job | | | | | | | | | |
| Researcher[b*] | 5 | 144 | 125 | 86.8 | 80.3 | 91.4 | Reference | | |
| Graduate student[b*] | 4 | 110 | 83 | 75.5 | 66.6 | 82.6 | 0.409 | 0.153 | 1.094 |
| Surveillance[b] | 5 | 165 | 141 | 85.5 | 79.3 | 90.0 | 0.894 | 0.343 | 2.328 |
| Lab technician | 4 | 115 | 109 | 94.8 | 89.1 | 97.6 | 2.915 | 0.869 | 9.779 |
| Undergraduate | 9 | 290 | 192 | 66.2 | 60.6 | 71.4 | **0.240** | **0.105** | **0.548** |
| Basic training | | | | | | | | | |
| Biology[b] | 19 | 567 | 459 | 81.0 | 77.5 | 84.0 | Reference | | |
| Biomedicine | 3 | 92 | 77 | 83.7 | 74.8 | 89.9 | 1.302 | 0.365 | 4.642 |
| Veterinary medicine | 2 | 66 | 40 | 60.6 | 48.6 | 71.5 | 0.260 | 0.064 | 1.053 |
| Nursery | 1 | 33 | 24 | 72.7 | 55.8 | 84.9 | 0.491 | 0.071 | 3.415 |
| High school | 2 | 66 | 50 | 75.8 | 72.0 | 90.7 | 0.643 | 0.153 | 2.698 |
| Institution | | | | | | | | | |
| University | 15 | 477 | 355 | 74.4 | 70.3 | 78.1 | Reference | | |
| Reference laboratory[b] | 7 | 182 | 154 | 84.6 | 78.7 | 89.1 | 1.833 | 0.718 | 4.681 |
| Surveillance | 5 | 165 | 141 | 85.5 | 79.3 | 90.0 | 2.198 | 0.772 | 6.259 |
| Total | 27 | 824 | 650[c] | 78.9 | 76.0 | 81.5 | - | - | - |

[a] Odds ratios from bivariate generalized linear mixed models (binomial error distribution, logit link-function) with user ID and bug species specified as random effects; odds ratios whose 95% CI does not include 1.0 are highlighted in bold typeface.

[b] Variables used in modeling (see hypotheses in Table 3).

[c] Of the 174 wrong identifications, 158 led to the identification of the wrong species, and 16 to no identification.

* The classes 'Researcher' and 'Graduate student' were merged into a single class ('Academic research/teaching') for multivariate modeling (see also Table 1).

CI-low/CI-up, lower/upper limits of the 95% confidence interval.

TRIATODEX performance was substantially larger among triatomine-bug species (random-effect standard deviation [SD] 0.915, CI 0.629–1.333) than among TRIATODEX users (SD 0.498, CI 0.285–0.869) (S2 Fig). The approximate CIs of species-specific random-effect conditional mode values spanned only negative values for *Triatoma guazu*, *Rhodnius prolixus*, *Psammolestes tertius*, *T. matogrossensis*, *T. rubrovaria*, and *R. pictipes*, suggesting that they were consistently harder to identify than average (S2 Fig).

Finally, our preliminary comparison of TRIATODEX and Galvão and Dale's printed key [2] suggested that, although both perform equally well, they differ in the average time required to complete an identification task. In particular, a logit-binomial GLMM with random-intercept terms for user-ID and bug-species-ID and a TRIATODEX fixed effect estimates a near-zero TRIATODEX effect (OR 1.089, CI 0.614–1.933), and hence predicts similar frequencies of correct identification (86.01%, CI 68.80–94.49 for TRIATODEX; 84.95%, CI 67.05–93.99 for [2]). This model also has a larger AICc score (by 1.96 units) than a 'null', random-effects only model–

**Table 5. TRIATODEX performance: Importance of variables selected for hypothesis testing and their model-averaged, adjusted effect estimates.**

| Trait | Importance[a] | Odds ratio | CI lower | CI upper |
|---|---|---|---|---|
| User traits[b] | | | | |
| Specialized training | 1.00 | 5.959 | 3.094 | 11.477 |
| Gender (female) | 0.50 | 0.659 | 0.374 | 1.161 |
| Age (over 40) | 0.48 | 0.602 | 0.294 | 1.233 |
| Reference laboratory | 0.40 | 1.477 | 0.732 | 2.980 |
| Academic research/teaching | 0.38 | 0.712 | 0.356 | 1.426 |
| Basic training in biology | 0.28 | 0.906 | 0.477 | 1.719 |
| Surveillance | 0.27 | 0.940 | 0.411 | 2.152 |
| Bug-species traits[b] | | | | |
| Tribe Rhodniini | 0.45 | 0.575 | 0.251 | 1.316 |
| Native-domestic | 0.28 | 1.208 | 0.425 | 3.435 |

[a] Variable importance computed as the sum of Akaike weights across the models containing each variable [47].

[b] Traits ranked by importance within each category.

CI-lower/CI-upper, lower/upper limits of the 95% confidence interval.

which suggests that, on average, TRIATODEX and the printed key both yielded correct identifications in about 85.48% of the tasks (CI 68.89–93.99%). Zero-truncated negative-binomial time models, in contrast, clearly suggested that the average time taken to complete an identification task was shorter when using TRIATODEX (4.72 min, CI 3.84–5.82) than when using the printed key (7.03 min, CI 5.73–8.62). This TRIATODEX-effect time model (S3 Table) overwhelmingly outperformed (AICc 42.56 units smaller) the corresponding 'null' model with only random effects; further, adding a factor indexing tasks that led to correct *vs.* incorrect species identifications yielded no improvement (AICc 1.35 units larger) and a near-zero difference estimate (rate ratio correct: incorrect 0.935, CI 0.801–1.092). Overall, then, TRIATODEX and Galvão and Dale's printed key [2] performed similarly, yet bug identification was, on average, 32.8% (CI 24.7–40.1%) faster with TRIATODEX–for a mean absolute saving of ~2.3 minutes per identification task.

## Discussion

TRIATODEX is a pictorial, annotated, polytomous electronic key to the Triatominae–the subfamily of blood-feeding bugs to which Chagas disease vectors belong [1,16]. In its current version, TRIATODEX runs on Android-based smartphones; a version compatible with both Android and iOS operation systems is under development. Our assessment of TRIATODEX performance yielded encouraging results. First, correct-identification probabilities, whether observed (Table 4) or model-predicted (Figs 2 and 3), were overall high. Second, TRIATODEX performance was particularly good in the hands of users with specialized training in triatomine-bug taxonomy, but varied little with other traits (of users and bugs) that we had hypothesized might affect it (Tables 3 and 5). Third, random-effect estimates suggested that variation in TRIATODEX performance (above and beyond that explained by the traits in Table 3) was larger among bug species than among users (S2 Fig). Finally, TRIATODEX performance matched that of a standard, pictorial printed key in a pilot trial, yet identification tasks were, on average, 33% less time (S3 Table).

### Taxonomic novelty and taxonomic coverage

TRIATODEX is the only available key covering virtually all (see below and S1 Table for the few exceptions) known species of Triatominae. Updating or expanding printed keys [1,2,34] to

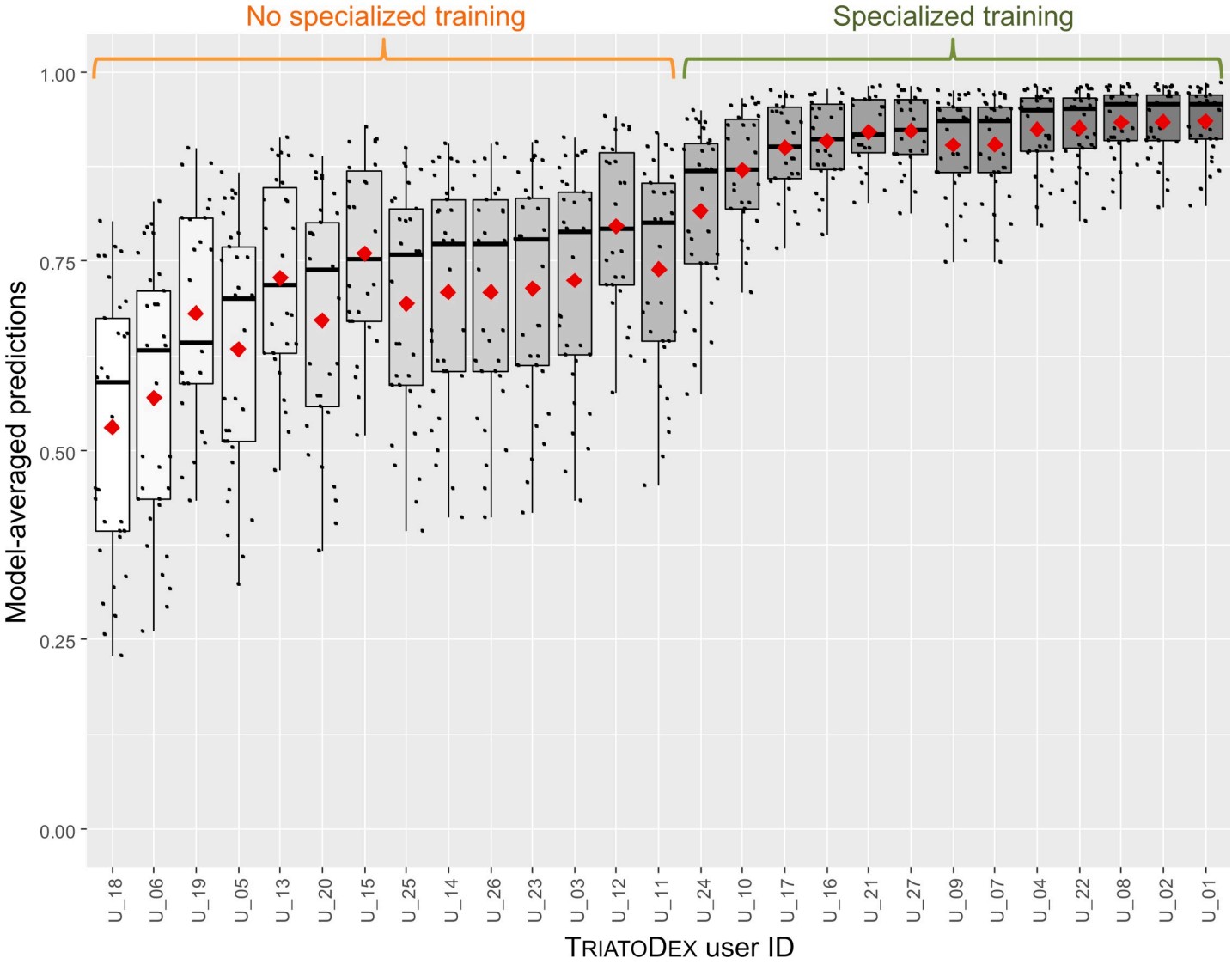

**Fig 2. Predicted proportions of correct triatomine-bug identification (33 species) by 27 TRIATODEX users.** Predictions are derived using Akaike weights from a set of 512 generalized linear mixed models; note that users with specialized training in triatomine-bug taxonomy did clearly better than those without. Black dots are user- and species-specific predictions; boxplots show medians (thick horizontal lines), quartiles (boxes), and values that fall within 1.5 times the interquartile range (whiskers); red diamonds are means.

increase taxonomic coverage is intrinsically difficult–they would have to be periodically revised and either reprinted or made available in electronic-document format (e.g., PDF). TriatoKey [10], an electronic key, would of course be easier to update; it now includes a subset of 42 triatomine-bug species recorded in Brazil, and would therefore have to incorporate 100 + further taxa to reach full coverage. TRIATODEX combines near-full taxonomic coverage (S1 Table) with easy integration of taxonomic novelty–which arises every so often as, for example, new species are described, already-described species are synonymized, or once-recognized species are revalidated. In the Triatominae, recently described species include *Mepraia parapatrica* [49], *Triatoma jatai* [50], *Rhodnius barretti* [36], *R. montenegrensis* [37], or *R. marabaensis* [38]. *Triatoma bahiensis* was revalidated after having been synonymized with *T. lenti* [51], and *R. taquarussuensis* was described as distinct from *R. neglectus* in 2016 [52] but

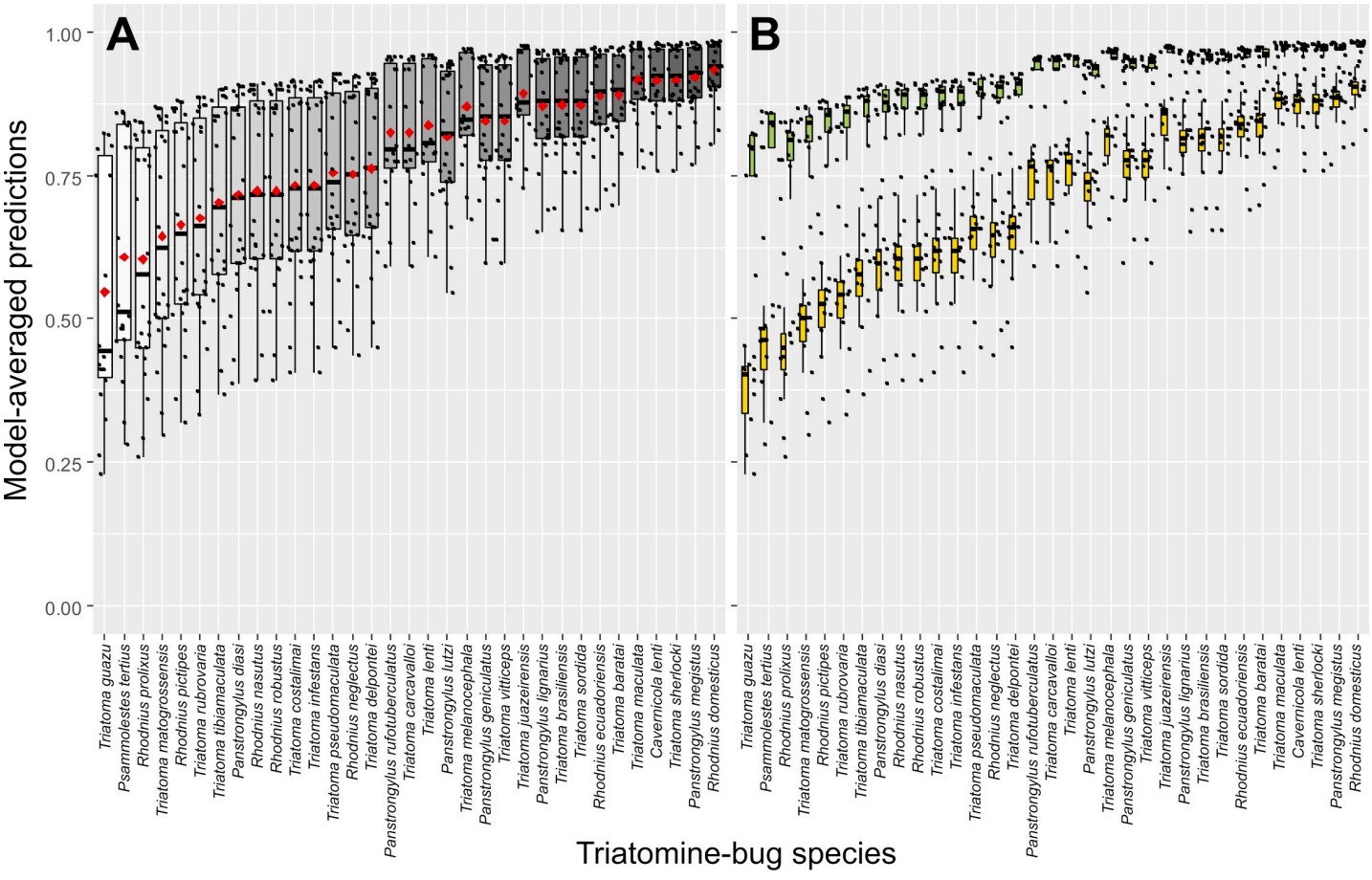

**Fig 3. Predicted proportions of TRIATODEX-based correct identification (by 27 users) across 33 triatomine-bug species.** Predictions are derived using Akaike weights from a set of 512 generalized linear mixed models. **A**, overall plot; in **B**, values are plotted separately for users with (green boxplots) and without (orange boxplots) specialized training in triatomine-bug taxonomy; note that users with specialized training did consistently better than those without across species. Black dots are species- and user-specific predictions; boxplots show medians (thick horizontal lines), quartiles (boxes), and values that fall within 1.5 times the interquartile range (whiskers); red diamonds are means.

synonymized back with it in 2019 [53]. Finally, *Nesotriatoma confusa* [54] was described based on material mistaken for *N. bruneri*, which was then synonymized with *N. flavida* [54], and *T. mopan* [55] and *T. huehuetenanguensis* [56] were described within the *T. dimidiata* complex [16]. Except for these two new *Triatoma* species [55,56] and the changes involving *R. taquarus-suensis* [52,53] and *Nesotriatoma* spp. [54], the version of TRIATODEX evaluated here covers all this taxonomic novelty–and the Android/iOS-compatible version currently under development will have full coverage (see S1 Table).

## Performance and comparison with other identification tools

TRIATODEX performed well in a series of 824 blind identification tasks involving 33 triatomine-bug species and 27 users. We are unaware of any similar assessment of how triatomine-bug taxonomic keys perform, but several alternative identification methods have been evaluated [35,57–61]. Because they tackle multiple-species problems similar to the one considered here, we did a detailed comparison of our results and those of [35] and [57]. We note that one set-back of these studies is that the key metric they report (percent 'success rate') lacks any measure of uncertainty–and also that computing overall 'success rates' by averaging over other

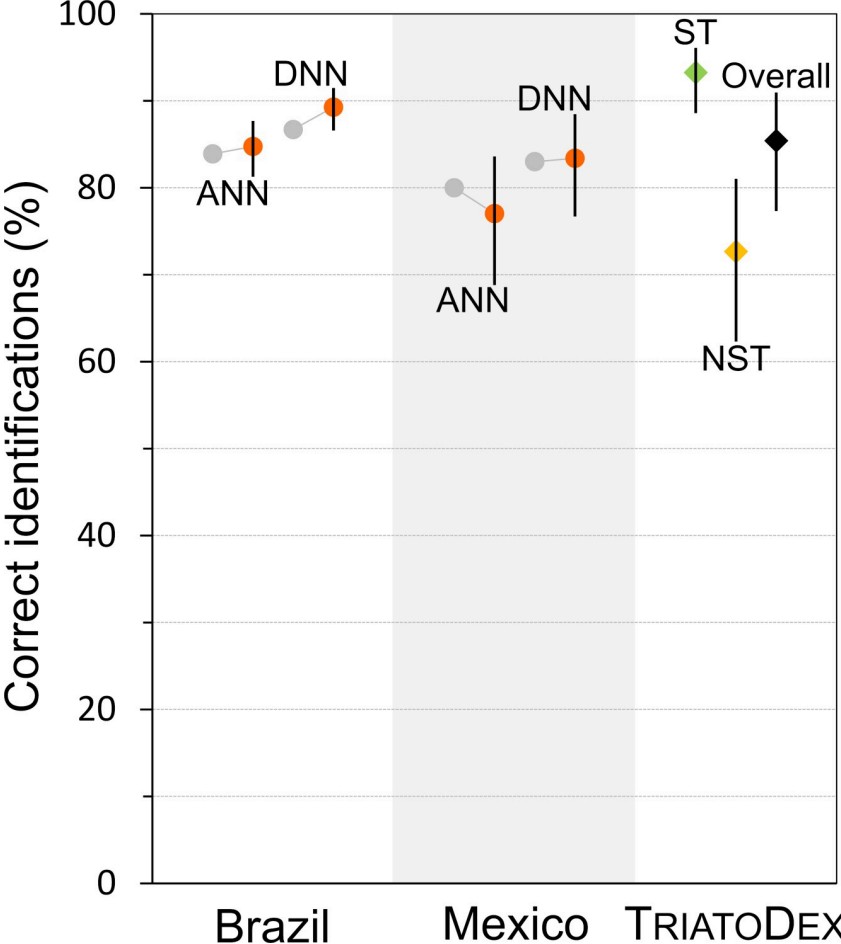

**Fig 4. TRIATODEX performance: A comparison with neural network-based models used in automated triatomine-bug identification [35,57].** We used data provided in [57] and present them either as reported (grey circles) or after a reanalysis based on generalized linear mixed models (red circles) similar to the ones we used to assess TRIATODEX performance (diamonds). The data in [57] refer to Brazilian and Mexican triatomines identified using either artificial neural networks (ANN, [35]) or deep neural networks (DNN, [57]). TRIATODEX results refer to the top-ranking model in our analyses (S2 Table), which distinguishes users with specialized training in triatomine-bug taxonomy (ST, green diamond) from those without (NST, orange diamond); the last value (Overall, black diamond) refers to a 'null' model without fixed-effect covariates. Error bars are 95% confidence intervals; note that the 'success rates' reported in [57] (grey circles) lack any measure of uncertainty. The values are given as percentages to match [35,57].

percentages (say, species-specific 'success rates') is "fundamentally flawed" ([62], p. 261). These caveats apply, in any case, to other, analogous evaluations based on crude 'success' proportions or percentages (e.g., [63,64]).

Khalighifar et al. [57] report the following 'success rates': for Brazilian bugs, 83.9% with the artificial neural networks (ANNs) of ref. [35] and 86.7% with the deep neural networks (DNNs) of ref. [57]; and, for Mexican bugs, 80.3% with ANNs and 83.0% with DNNs. To make these results more directly comparable with ours, we reanalyzed Khalighifar et al.'s data (Appendices I and II in [57]; see S3 Data) using a GLMM with a species-ID random effect and classifier (ANN or DNN) and region (Brazil or Mexico) fixed effects (S3 Code). The results (Fig 4) show that (i) TRIATODEX performance was, on average, similar to that of the highly sophisticated neural-network classifiers; when in the hands of our expert volunteers, TRIATO-DEX performed even better–and nonexpert users also did reasonably well; (ii) uncertainty, as

represented by CIs, was nontrivial, yet it went unmeasured in [35] and [57]; and (iii) there was some bias in the crude 'success rates' reported in ref. [57], perhaps as a result of averaging over species-specific percentages [62]. Finally, we observe that the 'success rates' of both ANNs and DNNs improved substantially when the algorithms worked on small subsets of 2–12 'candidate species' predicted to be present at the collection site of each problem specimen [35,57]; not surprisingly, the smaller the candidate-species subset, the higher the success rate, with values consistently above 95.0% and up to 99.4% for two-species subsets [35,57]. We did not try this 'faunal-subset' approach with our volunteers, but have all reasons to believe that it could bring similar improvements to TriatoDex performance. Note that, in our trials, users did not have access to the sites of origin of the bugs, and therefore were not able to use the distribution data provided in TriatoDex; this would probably have reduced the frequency of wrong identifications–at least for the 27 instances (15.5% of all wrong identifications) in which 16 species not present in South America were 'identified' in our sample of South American species.

## Pilot comparison with a printed key

We did a pilot trial comparing the performance of TriatoDex and a printed, pictorial key to 65 species of Triatominae [2]. The results were also encouraging, although we caution that they are to be regarded as preliminary (see also **Caveats** below). First, the odds of correct identification were fairly high, and effectively indistinguishable, for both keys; second, the average time taken to complete an identification task was reduced by one-third when using TriatoDex (S3 Table). This combination of good performance (as good as that of a standard printed key) and speed may be important when users have to identify large numbers of bugs, as is often the case in routine-surveillance systems and, on occasion, in research contexts. At the average rates we estimated (4.7 min/task for TriatoDex and 7.0 min/task for [2]), an average user would save nearly four hours of work for every 100 identification tasks completed. Once trained, automated classifiers such as ANNs and DNNs are much faster [35,57], and a system allowing users to upload bug pictures to a virtual server running one of those algorithms and returning an identification would be very useful.

## Caveats

First, it should be clear that some bugs will inevitably be misidentified with TriatoDex. As our comparative analyses show, however, this snag is not specific to our key: other bug-identification tools, from standard printed keys to sophisticated deep-learning algorithms, will fail at about the same rate. More experienced users and better algorithms will fail less often, particularly when aware of the subsets of 'candidate species' that occur at the sites where problem specimens were caught. The cost of misidentifying a bug may vary widely–from very high if it involves failing to detect the presence of dangerous non-native species, such as *T. infestans* in Brazil or *R. prolixus* in Central America [22], to near-zero if the species involved have similar degrees of medical relevance–e.g., *R. montenegrensis* and *R. marabaensis*. Misidentification can also result in wrong distribution records and thus confound both species-distribution modeling and the test of hypotheses about species-specific drivers of habitat suitability (e.g., [65]). The patterns of species misclassification in our assessment of TriatoDex performance show that, except for five tasks in which the bugs were not identified, wrong classifications by expert users involved same-genus species–and, in many cases, morphologically similar species such as, e.g., *Psammolestes tertius* and *Ps. arthuri* or *T. matogrossensis* and *T. williami* (see S4 Table). In contrast, misidentifications by non-expert users sometimes involved very distantly-related, and very dissimilar, species–including, e.g., species of *Triatoma* and *Rhodnius* or species of *Alberprosenia* and *Panstrongylus* (S5 Table).

In a few extreme cases, the characters that can be described in a key may be insufficient to distinguish one species from some similar-looking relative(s) [1,16]. When this happens, all the species involved are listed by TRIATODEX in the last step of the identification task; in the picture of each species, the user is warned that the bug may belong to the pictured species but, "considering basic external morphology", that species is "indistinguishable from" species 'X', 'Y', etc... In the current version of the key, this proviso affects (i) *Nesotriatoma flavida* and *N. bruneri* (which have in fact been synonymized [54]); (ii) *Rhodnius robustus* (*s.l.*), *R. montenegrensis* (formerly *R. robustus* II [66]), *R. marabaensis* (likely *R. robustus* III [16]), and *R. barretti* (although careful evaluation might reveal diagnostic characters [36]); (iii) *R. neglectus*, *R. milesi* (most likely a synonym of *R. neglectus* [16]), and *R. taquarussuensis* (recently synonymized with *R. neglectus* [53]); and (iv) *R. domesticus* and *R. zeledoni* (which are probably synonyms [16]). The distribution maps and ecological notes available in TRIATODEX may help in some cases, but more sophisticated tools such as DNA sequencing or morphometrics may be needed in others [16,61]. Along these same lines, note that immature stages (eggs and nymphs) are not covered in TRIATODEX; the usual advice here is to rear field-caught immatures to adulthood in the laboratory and then identify them [1].

We stress that, strictly speaking, the results of the assessment of TRIATODEX performance we report here only apply to the set of users described in Table 1, the set of bug species listed in Table 2, and the set of user and bug traits given in Table 3. Although users had a variety of backgrounds and levels of experience, test species covered three tribes and five genera, and both user and bug traits were hypothesis-based, none of those sets is a representative, probabilistic sample of its parent population–the sets of, respectively, all potential TRIATODEX users, all triatomine-bug species, and all user and bug traits possibly affecting identification success. Readers should keep this in mind when interpreting our findings.

Finally, it is clear that, in real-world practice, a reference-standard identification for each specimen will rarely, if ever, be available. In general, therefore, identification-task results are best regarded as *hypotheses* whose plausibility has to be gauged against knowledge on the systematic affinities (in particular for bugs in morphologically similar species), phenotypic variability, distribution, ecology, or behavior of each species. As we have shown here, specific training in triatomine-bug taxonomy is likely to lead, on average, to more plausible hypotheses.

## Prospects

TRIATODEX needs to be updated to cover the latest information on triatomine-bug taxonomy, including the description of *N. confusa* (and the synonymization of *N. bruneri* with *N. flavida* [54]) or the synonymization of *R. taquarussuensis* with *R. neglectus* [53]. We plan to schedule an annual round of updating to incorporate taxonomic novelty on a regular basis. Updating will also allow for gradually refining distribution records and maps (which are currently very rough) and the descriptions of characters and character states; other improvements, such as incorporating more and, in some cases, better-quality pictures as they become available, are also contemplated.

Our assessment of TRIATODEX performance pinpointed some species that appear to be consistently harder to identify than average (S2 Fig); further research is needed to (i) see if the same is true for other species not tested so far, (ii) discover whether the precise sources of difficulty relate to app features (e.g., low-quality pictures or ambiguous character-state descriptions), and (iii) find and test characters or character states with the potential to ease identification. A similar, careful search for diagnostic characters and character states could help distinguish *R. barretti* [36] from the (relatively distantly-related [16,36]) members of the

*R. robustus-R. prolixus* complex. Within that complex, it might perhaps be possible to tell apart *R. robustus* (*s.l.*) from the recently described *R. montenegrensis* and *R. marabaensis* [37,38]; more accurate descriptions of the known distribution ranges would also help in this case–and probably in others. Finally, incorporating intraspecific phenotypic variation, perhaps through richer descriptions and/or multiple pictures (e.g., a small 'photo gallery' for each species), would probably help improve TRIATODEX performance for at least some species–such as, for example, the phenotypically variable *T. infestans*, *T. brasiliensis*, *T. dimidiata*, *T. rubrovaria*, *T. protracta*, *Mepraia* spp., *P. geniculatus*, or *R. ecuadoriensis* [1,16,34,67–71].

Two more challenges lie ahead. One is simple enough and already virtually solved–developing and releasing an iOS/Android-compatible, updated version of the app. The other will likely be much tougher; it involves designing and completing a trial to see whether and to what extent TRIATODEX would be useful in real-world Chagas disease vector control-surveillance. For example, surveillance teams from different municipalities could be randomized to use either a standard, printed key as per current routine practice (the 'control' group) or TRIATO-DEX (the 'treatment' group), and 'treatment effects' could be measured in terms of pre-specified metrics such as percent correct identifications, time spent in identification tasks, or operational costs. One potentially interesting extension of such a trial would be to use TRIATO-DEX as a means of engaging community-health agents and other primary healthcare staff in triatomine-bug and, thereby, Chagas disease surveillance. Further extension to the broader community would require that the new version of the app places more emphasis on the distinction between triatomine bugs and other insects of similar appearance and commonly found in/around houses–including non-blood-feeding bugs and some beetles or cockroaches. This would simply entail adding a few extra pictures and a brief, clear description of key, easy-to-check morphological characters other than the rostrum (e.g., [72]) in the opening steps of the 'Search' option (Fig 1).

## Conclusions

TRIATODEX holds much promise as a handy, flexible, and fairly reliable tool for triatomine-bug identification; it has near-full taxonomic coverage and performs at least as well as standard printed keys and sophisticated neural-network models. An updated iOS/Android version is under development. Our analyses suggest that, as TRIATODEX results accrue to cover more taxa, they may help pinpoint triatomine-bug species that are consistently harder (than average) to identify. We expect that, with continuous refinement derived from evolving knowledge and user feedback (which can be provided through the 'Contact' option of the app's main menu), TRIATODEX will substantially help strengthen both routine entomological surveillance and eco-epidemiological research on Chagas disease vectors.

## Supporting information

**S1 Fig. Correlation matrix and variance inflation factors (VIFs; red font) for variables used in the assessment of TriatoDex performance (see Table 3 of the main text).** Note that correlations were all ≤ |0.47| and VIFs were all < 2.0.
(TIF)

**S2 Fig. Random-effect conditional modes (with 95% confidence intervals, CI) for 33 triatomine-bug species and 27 TriatoDex users; values derived from the top-performing model (S2 Table).** Note that random variation in TRIATODEX performance was substantially larger among species than among users; the CIs spanned only negative values for six bug species but

only for one user.
(TIF)

**S1 Table. Triatomine-bug taxa covered by the current version of TriatoDex.**
(XLSX)

**S2 Table. Top-performing model (binomial, logit link-function) of TriatoDex performance: Parameter estimates, standard errors, and 95% confidence interval limits.**
(PDF)

**S3 Table. Top-performing time model (zero-truncated negative binomial, log link-function) of TriatoDex performance in comparison with a printed key [2]: Parameter estimates, standard errors, and 95% confidence interval limits.**
(PDF)

**S4 Table. Patterns of species misclassification by TriatoDex users with specialized training in triatomine-bug taxonomy.**
(XLSX)

**S5 Table. Patterns of species misclassification by TriatoDex users without specialized training in triatomine-bug taxonomy.**
(XLSX)

**S1 Data. Data underlying the assessment of overall TriatoDex performance.**
(TXT)

**S2 Data. Data underlying the assessment of TriatoDex performance in comparison with a printed key [2].**
(TXT)

**S3 Data. Data underlying the assessment of performance of the neural-network models in refs. [35,57].**
(TXT)

**S1 Code. R code used to assess overall TriatoDex performance.**
(TXT)

**S2 Code. R code used to assess TriatoDex performance in comparison with a printed key [2].**
(TXT)

**S3 Code. R code used to assess the performance of the neural-network models in refs. [35,57].**
(TXT)

## Acknowledgments

We are grateful to Carolina Dale and Cléber Galvão for useful suggestions, to the Fiocruz Collection of Chagas Disease Vectors (Fiocruz-COLVEC) for specimen loans, and to all TRIATODEX users for participating in the trials. Cléber Galvão, James S Patterson, and José M Ayala generously provided some of the triatomine pictures used in TRIATODEX.

## Author Contributions

**Conceptualization:** Rodrigo Gurgel-Gonçalves, Fernando Abad-Franch, Douglas de Almeida Rocha.

**Data curation:** Rodrigo Gurgel-Gonçalves, Fernando Abad-Franch.

**Formal analysis:** Rodrigo Gurgel-Gonçalves, Fernando Abad-Franch.

**Funding acquisition:** Rodrigo Gurgel-Gonçalves.

**Investigation:** Rodrigo Gurgel-Gonçalves, Fernando Abad-Franch, Maxwell Ramos de Almeida, Marcos Takashi Obara, Rita de Cássia Moreira de Souza, Jainaine Abrantes de Sena Batista, Douglas de Almeida Rocha.

**Methodology:** Rodrigo Gurgel-Gonçalves, Fernando Abad-Franch, Maxwell Ramos de Almeida, Marcos Takashi Obara, Rita de Cássia Moreira de Souza, Jainaine Abrantes de Sena Batista, Douglas de Almeida Rocha.

**Project administration:** Rodrigo Gurgel-Gonçalves.

**Resources:** Rodrigo Gurgel-Gonçalves.

**Supervision:** Rodrigo Gurgel-Gonçalves, Maxwell Ramos de Almeida, Marcos Takashi Obara, Rita de Cássia Moreira de Souza, Jainaine Abrantes de Sena Batista, Douglas de Almeida Rocha.

**Validation:** Rodrigo Gurgel-Gonçalves, Fernando Abad-Franch, Maxwell Ramos de Almeida, Marcos Takashi Obara, Rita de Cássia Moreira de Souza, Jainaine Abrantes de Sena Batista, Douglas de Almeida Rocha.

**Visualization:** Fernando Abad-Franch.

**Writing – original draft:** Rodrigo Gurgel-Gonçalves, Fernando Abad-Franch.

**Writing – review & editing:** Rodrigo Gurgel-Gonçalves, Fernando Abad-Franch, Maxwell Ramos de Almeida, Marcos Takashi Obara, Rita de Cássia Moreira de Souza, Jainaine Abrantes de Sena Batista, Douglas de Almeida Rocha.

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
