## [Decision Letter · Decision Letter 0]

16 Dec 2020

PONE-D-20-27928

TriatoDex, an electronic identification key to the Triatominae: development, description, and performance

PLOS ONE

Dear Dr. Gurgel-Gonçalves,

Thank you for submitting your manuscript to PLOS ONE. After careful consideration, we feel that it has merit but does not fully meet PLOS ONE’s publication criteria as it currently stands. Therefore, we invite you to submit a revised version of the manuscript that addresses the points raised during the review process. We received feedback from reviewers and both were positive about your manuscript. At the same time, both raised very relevant points about specific aspects in the manuscript. In this way, I decided to characterize it as "major reviews", but I believe that the suggestions are not too complex to be addressed.

I look forward to a new version of the manuscript to continue the evaluation.

We look forward to receiving your revised manuscript.

Kind regards,

Sergio N. Stampar, Dr.

Academic Editor

PLOS ONE

Journal Requirements:

3. Thank you for stating the following in the Competing Interests/Financial Disclosure* (delete as necessary) section:

We note that one or more of the authors are employed by a commercial company: H2J Comunicação & Marketing, Brasília, Brazil

(2) Please also provide an updated Competing Interests Statement declaring this commercial affiliation along with any other relevant declarations relating to employment, consultancy, patents, products in development, or marketed products, etc.  

Please respond by return email with an updated Funding Statement and Competing Interests Statement and we will change the online submission form on your behalf.

Reviewers' comments:

Reviewer's Responses to Questions

**Comments to the Author**

1. Is the manuscript technically sound, and do the data support the conclusions?

Reviewer #1: Yes

Reviewer #2: Yes

2. Has the statistical analysis been performed appropriately and rigorously? 

Reviewer #1: I Don't Know

Reviewer #2: Yes

3. Have the authors made all data underlying the findings in their manuscript fully available?

Reviewer #1: Yes

Reviewer #2: Yes

4. Is the manuscript presented in an intelligible fashion and written in standard English?

Reviewer #1: Yes

Reviewer #2: Yes

5. Review Comments to the Author

Reviewer #1: Review Plos Triatodex

I have a few doubts on the importance of this manuscript, as listed below:

1-The manuscript brings the analysis of a very important technological tool for the Triatominae identification and the detailed statistical analysis checking the efficacy of the Triatodex App. Despite the nice presentation of the manuscript, I am not sure it is in the scope or interesting to be published in Plos one. Therefore, I am going to decline of presenting the final decision which I leave with the editorial team.

2-Another point is that the tool structure, tutorial, and the program itself are available on the internet since 2017 and I am not sure if this manuscript is going to complement any previous information already available in the web since 2017.

3-How important is the publication of the statistical analysis on the efficacy of the Triatominae identification tool? Its importance and efficiency are going to be proved or disapproved by the users themselves, assessing and using the App. Again, the final decision I am going to leave with the editors.

Please check the following links presenting the Triatodex in the internet since 2017

https://www.youtube.com/watch?v=8-M55EoWjg0

https://www.youtube.com/watch?v=8jqGNnkU5eU

https://blogdeparasitologia.wordpress.com/2017/09/22/aplicativo-facilita-a-identificacao-morfologica-de-triatomineos/

Finaly, this manuscript does not bring complementary information or bring crucial updates related to the App. Several other papers on Triatominae systematics and taxonomy could be included to enrich the knowledge of the more than 150 species now.

Some suggestions are listed below:

-Title should include further information such as: the Trypanosoma cruzi vectors, Hemiptera, Heteroptera, Reduviidae, Triatominae

-Additional references on Triatominae systematics and taxonomy should be added in order to complement the visual and summarized information for each of the listed species in the App.

Why just Lent and Wygodzinsky, Carcavalo et al and Galvão & Dale were mentioned? And about the extensive bibliography on the species complexes? Phenotipic variations? Why these papers were not mentioned and listed?

Reviewer #2: The authors present a novel and useful digital tool for Triatomine species identification, the vectors of Trypanosoma cruzi, a parasite that causes Chagas Disease. TriatoDex is a cell phone app currently available for android phones, which will be available also for iPhones. The authors also performed several tests to assess the performance of TriatoDex in different conditions, such as User training background, user age, insect identification difficulty, etc., in a hypothesis testing framing.

Overall, I think this is a very nice project (TriatoDex), which has been creatively evaluated under different situations and compared to other ways to identify species, such as traditional keys or automated approaches. Triatodex outperformed in several of such challenges.

I can realize this project is the result of hard work, possibly of many years. I can testify the author's group includes well-established experts in Triatomines. I encourage authors to keep TriatoDex updated and expanded when more resources and applications emerge. I appreciate the Prospect section, where authors envisage ways to improve app performance.

My only concern about the models is that the user ID was used as a random variable. Still, in my opinion, tasks are nested in users (i.e., identification tasks are not independent observations). If true, a nested modeling approach could be more suitable for this data.

Minor points:

Line 5.- Consider replacing "Triatominae" with "Insect vectors."

Line 10-.- Add "worldwide" before 150

Line 28. Any clue to the next steps to improve the reliance of TriatoDex to be used by not trained users?

Line 51.- It may be of help to better understand the Chagas disease dimensions, adding a few reasons of why Chagas disease is the main parasitic disease in Latin America

Line 76: Rephrase to emphasize that some regions are species-rich, challenging vector control measures.

Line 85: Consider to add "worldwide" to make clear that TriatDex deals with the worldwide species diversity instead of regional coverage

Line 200: Please explain in more detail why 511 models

6. PLOS authors have the option to publish the peer review history of their article (what does this mean?). If published, this will include your full peer review and any attached files.

Reviewer #1: No

Reviewer #2: No

---

## [Author Response · Author response to Decision Letter 0]

23 Dec 2020

Reviewer #1: Review Plos Triatodex

I have a few doubts on the importance of this manuscript, as listed below:

1-The manuscript brings the analysis of a very important technological tool for the Triatominae identification and the detailed statistical analysis checking the efficacy of the Triatodex App. Despite the nice presentation of the manuscript, I am not sure it is in the scope or interesting to be published in Plos one. Therefore, I am going to decline of presenting the final decision which I leave with the editorial team.

We thank the Reviewer for her/his positive comments. As per PLOS ONE’s guidelines, available at https://journals.plos.org/plosone/static/publish and more specifically at https://journals.plos.org/plosone/s/submission-guidelines#loc-methods-software-databases-and-tools, we are confident that our submission is within the scope of the journal and meets its publication criteria (https://journals.plos.org/plosone/s/journal-information#loc-criteria-for-publication).

2-Another point is that the tool structure, tutorial, and the program itself are available on the internet since 2017 and I am not sure if this manuscript is going to complement any previous information already available in the web since 2017.

We provide a detailed description of the app and a rigorous evaluation of its performance. We also present the current limitations of our tool and outline some key perspectives for further development and improvement. We believe that sharing these topics with the broader community is important and falls well within the scope of PLOS ONE. Specifically, the journal welcomes “[S]ubmissions describing methods, software, databases, or other tools that meet the journal’s criteria for utility, validation and availability.” We believe our work fits that description and fulfils all three criteria (see https://journals.plos.org/plosone/s/submission-guidelines#loc-methods-software-databases-and-tools).

3-How important is the publication of the statistical analysis on the efficacy of the Triatominae identification tool? Its importance and efficiency are going to be proved or disapproved by the users themselves, assessing and using the App. Again, the final decision I am going to leave with the editors.

We note again that PLOS ONE’s guidelines state that “[S]ubmissions presenting methods, software, databases, or tools must demonstrate that the new tool achieves its intended purpose. If similar options already exist, the submitted manuscript must demonstrate that the new tool is an improvement over existing options in some way. This requirement may be met by including a proof-of-principle experiment or analysis; if this is not possible, a discussion of the possible applications and some preliminary analysis may be sufficient.” Our assessment of TriatoDex performance, including a comparison with a standard printed key, addresses this requirement.

Please check the following links presenting the Triatodex in the internet since 2017

https://www.youtube.com/watch?v=8-M55EoWjg0

https://www.youtube.com/watch?v=8jqGNnkU5eU

https://blogdeparasitologia.wordpress.com/2017/09/22/aplicativo-facilita-a-identificacao-morfologica-de-triatomineos/

We see our manuscript as a technical/scientific counterpart to the non-technical information these links provide access to. If our tool is to be used in disease-vector surveillance, the decision to do so will likely be based on the data and analyses we present in this manuscript.

Finaly, this manuscript does not bring complementary information or bring crucial updates related to the App. Several other papers on Triatominae systematics and taxonomy could be included to enrich the knowledge of the more than 150 species now.

Please see our previous comments. We have tried to reasonably cover the vast literature on triatomine-bug systematics; our emphasis on taxonomic novelty (refs. [36-38,49-56]) is, we believe, suitably counterbalanced by the inclusion of both classic (e.g., refs. [1,34]) and more recent in-depth reviews (e.g., refs. [2,16]). We note that we already have 67 refs.

Some suggestions are listed below:

-Title should include further information such as: the Trypanosoma cruzi vectors, Hemiptera, Heteroptera, Reduviidae, Triatominae

Perhaps “TriatoDex, an electronic identification key to the Triatominae (Hemiptera: Reduviidae), vectors of Chagas disease: development, description, and performance”

-Additional references on Triatominae systematics and taxonomy should be added in order to complement the visual and summarized information for each of the listed species in the App.

Why just Lent and Wygodzinsky, Carcavalo et al and Galvão & Dale were mentioned? And about the extensive bibliography on the species complexes? Phenotipic variations? Why these papers were not mentioned and listed?

Please see our comment above. We note that those three refs. include the most widely-used printed identification keys to the Triatominae; that is why we specifically mention them.

Reviewer #2: The authors present a novel and useful digital tool for Triatomine species identification, the vectors of Trypanosoma cruzi, a parasite that causes Chagas Disease. TriatoDex is a cell phone app currently available for android phones, which will be available also for iPhones. The authors also performed several tests to assess the performance of TriatoDex in different conditions, such as User training background, user age, insect identification difficulty, etc., in a hypothesis testing framing.

Overall, I think this is a very nice project (TriatoDex), which has been creatively evaluated under different situations and compared to other ways to identify species, such as traditional keys or automated approaches. Triatodex outperformed in several of such challenges.

I can realize this project is the result of hard work, possibly of many years. I can testify the author's group includes well-established experts in Triatomines. I encourage authors to keep TriatoDex updated and expanded when more resources and applications emerge. I appreciate the Prospect section, where authors envisage ways to improve app performance.

We thank the Reviewer for these positive remarks.

My only concern about the models is that the user ID was used as a random variable. Still, in my opinion, tasks are nested in users (i.e., identification tasks are not independent observations). If true, a nested modeling approach could be more suitable for this data.

The Reviewer is right – the results of tasks performed by the same user cannot be treated as independent. Note, however, that there is only one ‘lower level’ below the ‘higher level’ represented by users – observations nested within users are the lowest possible level of the hierarchy. The structure of our models captures this hierarchy, but cannot go below its lowest level. That would be different if, for example, each user had undertaken several identification tasks involving bugs of the same species; then, one could argue that the problem should be represented by a three-level hierarchy with observations nested within species nested within users – which could perhaps be written as {user[species(outcomes)]}. In our study, each user had to identify just one bug of each species, so the hierarchy was simply [user(outcomes)] and the models capture this structure.

Minor points:

Line 5.- Consider replacing "Triatominae" with "Insect vectors."

We prefer to use Triatominae; “insect vectors” sounds too general.

Line 10-.- Add "worldwide" before 150

Changed to “…the 150 triatomine-bug species described worldwide…”

Line 28. Any clue to the next steps to improve the reliance of TriatoDex to be used by not trained users?

We have already 301 words in the Abstract; hurdles and ways for improvement are discussed in some detail within the main body of the manuscript, particularly in the ‘Caveats’ and ‘Prospects’ sections. The last sentences of the ‘Prospects’ section specifically address the extension of TriatoDex use to “… community-health agents and other primary healthcare staff…” and “… the broader community…”

Line 51.- It may be of help to better understand the Chagas disease dimensions, adding a few reasons of why Chagas disease is the main parasitic disease in Latin America

We have changed this to “… one of the major neglected tropical diseases …” The next paragraph presents recent estimates showing high prevalence, with millions infected, and high incidence, with thousands of new infections occurring each year.

Line 76: Rephrase to emphasize that some regions are species-rich, challenging vector control measures.

Perhaps “Particularly in species-rich regions, these native triatomines pose a dual challenge to Chagas disease surveillance”

Line 85: Consider to add "worldwide" to make clear that TriatDex deals with the worldwide species diversity instead of regional coverage

Changed to “… covers the 150 triatomine-bug species formally described worldwide up to 2017…”

Line 200: Please explain in more detail why 511 models

We have 9 variables; the number of possible additive combinations of those variables is 29 = 512, so we have the full model plus the “… 511 additive models nested within [that] full model…”.

---

## [Decision Letter · Decision Letter 1]

3 Mar 2021

TriatoDex, an electronic identification key to the Triatominae (Hemiptera: Reduviidae), vectors of Chagas disease: development, description, and performance

PONE-D-20-27928R1

Dear Dr. Gurgel-Gonçalves,

We’re pleased to inform you that your manuscript has been judged scientifically suitable for publication and will be formally accepted for publication once it meets all outstanding technical requirements.

Kind regards,

Sergio N. Stampar, Dr.

Academic Editor

PLOS ONE

Additional Editor Comments (optional):

Dear Authors

After a new evaluation, the reviewers indicated the acceptance of the manuscript. Still, one of the reviewers asked for attention at one point in the manuscript, see below. Please check this point in the final version.

"Some of the points were reasonably answered.

Unfortunately, the authors do not discuss on details about the phenotypic variation on triatomines. The authors very superficially mention on Lines 567 and 568 “– such as, for example, the phenotypically variable T. infestans, T. rubrovaria, T. protracta, Mepraia spp., P. geniculatus, or R. ecuadoriensis. No references were added!

Additionally, the most important Chagas disease vector of northeastern Brazil was neither mentioned nor referred. It is well known that T. brasiliensis presents an important phenotypic variation specially in the Pernambuco State. This information would be extremely helpful for the users of the Triatodex

These are my major concerns related to the manuscript."

Kind regards

Sergio

Reviewers' comments:

Reviewer's Responses to Questions

**Comments to the Author**

1. If the authors have adequately addressed your comments raised in a previous round of review and you feel that this manuscript is now acceptable for publication, you may indicate that here to bypass the “Comments to the Author” section, enter your conflict of interest statement in the “Confidential to Editor” section, and submit your "Accept" recommendation.

Reviewer #1: (No Response)

Reviewer #2: All comments have been addressed

2. Is the manuscript technically sound, and do the data support the conclusions?

Reviewer #1: Partly

Reviewer #2: Yes

3. Has the statistical analysis been performed appropriately and rigorously? 

Reviewer #1: I Don't Know

Reviewer #2: Yes

4. Have the authors made all data underlying the findings in their manuscript fully available?

Reviewer #1: Yes

Reviewer #2: Yes

5. Is the manuscript presented in an intelligible fashion and written in standard English?

Reviewer #1: Yes

Reviewer #2: Yes

6. Review Comments to the Author

Reviewer #1: Some of the points were reasonably answered.

Unfortunately, the authors do not discuss on details about the phenotypic variation on triatomines. The authors very superficially mention on Lines 567 and 568 “– such as, for example, the phenotypically variable T. infestans, T. rubrovaria, T. protracta, Mepraia spp., P. geniculatus, or R. ecuadoriensis. No references were added!

Additionally, the most important Chagas disease vector of northeastern Brazil was neither mentioned nor referred. It is well known that T. brasiliensis presents an important phenotypic variation specially in the Pernambuco State. This information would be extremely helpful for the users of the Triatodex

These are my major concerns related to the manuscript.

Reviewer #2: The authors did a good work addressing my comments and answering my critique about the app testing's statistical design. I have no further questions and recommend this paper for publication in PlosOne

7. PLOS authors have the option to publish the peer review history of their article (what does this mean?). If published, this will include your full peer review and any attached files.

Reviewer #1: No

Reviewer #2: No

---

## [Editor Report · Acceptance letter]

8 Apr 2021

PONE-D-20-27928R1 

TriatoDex, an electronic identification key to the Triatominae (Hemiptera: Reduviidae), vectors of Chagas disease: development, description, and performance 

Dear Dr. Gurgel-Gonçalves:

I'm pleased to inform you that your manuscript has been deemed suitable for publication in PLOS ONE. Congratulations! Your manuscript is now with our production department. 

Kind regards, 

on behalf of

Dr. Sergio N. Stampar 

Academic Editor

PLOS ONE